# Coffee Abundant in Chlorogenic Acids Reduces Abdominal Fat in Overweight Adults: A Randomized, Double-Blind, Controlled Trial

**DOI:** 10.3390/nu11071617

**Published:** 2019-07-16

**Authors:** Takuya Watanabe, Shinichi Kobayashi, Tohru Yamaguchi, Masanobu Hibi, Ikuo Fukuhara, Noriko Osaki

**Affiliations:** 1Biological Science Research Laboratories, Kao Corporation, 2-1-3 Bunka, Sumida, Tokyo 131-8501, Japan; 2Health Care Food Research Laboratories, Kao Corporation, 2-1-3 Bunka, Sumida, Tokyo 131-8501, Japan; 3Fukuhara Hospital, 3-1-15 Shimamatsuhigashimachi, Eniwa, Hokkaido 061-1351, Japan

**Keywords:** chlorogenic acids, instant coffee, randomized clinical trial, total fat area, visceral fat area

## Abstract

The components of roasted or green coffee beans that promote abdominal fat reduction are not clear. We investigated the effects of daily consumption of coffee enriched in chlorogenic acids (CGA) on abdominal fat area in a randomized, double-blind, parallel controlled trial. Healthy, overweight men and women (*n* = 150, body mass index (BMI) ≥25 to <30 kg/m^2^) were randomly allocated to high-CGA (369 mg CGA/serving) or control (35 mg CGA/serving) coffee groups. Instant coffee was consumed once daily for 12 weeks, with four-week pre- and post-observation periods. Abdominal fat area and anthropometric measurements were analyzed at baseline and at four, eight, and 12 weeks, and 142 subjects completed the trial. Visceral fat area (VFA), total abdominal fat area (TFA), body weight, and waist circumference significantly decreased in the CGA group compared with the control group, with a group × time interaction (*p* < 0.001, *p* = 0.001, *p* = 0.025, and *p* = 0.001, respectively). Changes in VFA and TFA from baseline to 12 weeks were significantly greater in the CGA group than in the control group (−9.0 ± 13.9 cm^2^ vs. −1.0 ± 14.3 cm^2^, *p* < 0.001; −13.8 ± 22.9 cm^2^ vs. −2.0 ± 16.2 cm^2^, *p* < 0.001). No severe adverse events occurred. Consumption of high-CGA coffee for 12 weeks by overweight adults might lower VFA, TFA, BMI, and waist circumference.

## 1. Introduction

Coffee is consumed globally since the 10th century and its frequent consumption continues in the modern world. Habitual coffee consumption has various beneficial effects on health. Many epidemiologic studies and meta-analyses assessed the relationship between coffee consumption and the risk of lifestyle-related diseases such as cardiovascular disease and diabetes [1,2,3,4,5,6,7]. A meta-analysis by Mostofsky et al. [8] including five epidemiologic studies involving a total of approximately 140,000 subjects revealed a significant positive correlation between drinking four or more cups of coffee per day and a decreased risk of cardiovascular disease. Coffee comprises many components with pharmacologic effects, such as caffeine [9,10], but the specific components associated with the beneficial effects of coffee consumption are unclear. Coffee beans contain chlorogenic acids (CGA), polyphenols that have anti-oxidant properties [11]. CGA is abundant in green coffee beans or eggplant exocarp. It is formed from an esterification reaction between caffeic acid or ferulic acid and quinic acid, and occurs in the following derivatives: 5-caffeoylquinic acid (5-CQA, formerly called 3-caffeoylquinic acid or chlorogenic acid) [12]; 3-caffeoylquinic acid (3-CQA); 4-caffeoylquinic acid (4-CQA); and their corresponding dimers, i.e., 3,4-dicaffeoylquinic acid (3,4-diCQA), 3,5-dicaffeoylquinic acid (3,5-diCQA), 4,5-dicaffeoylquinic acid (4,5-diCQA); 3-feruloylquinic acid (3-FQA); 4-feruloylquinic acid (4-FQA); and 5-feruloylquinic acid (5-FQA) [13,14]. Clinical trials demonstrated the various effects of CGA consumption on vascular endothelial function [15,16], blood pressure [17], sleep quality [18], and brain function [19]. While many clinical studies report beneficial effects of CGA on various physiologic functions, few studies assessed the effect of daily consumption of coffee containing high levels of CGA on body fat and weight in obese and/or overweight subjects. Some randomized clinical trials indicate that consuming high levels of CGA [20] or green coffee extract that contain abundant CGA [21] can reduce body weight and/or waist circumference (WC). The majority of clinical studies, however, did not compare the effects of CGA, caffeine, and a chemical coffee constituent with those of a conventional coffee beverage [22,23,24]. Moreover, the findings of a clinical study performed in 12 healthy volunteers drinking different coffee products containing CGA demonstrated that instant coffee enriched with CGA led to a reduction in body weight in the CGA-enriched and normal instant coffee groups [25]. Our previous study revealed that repeated consumption of pre-packaged caffeinated coffee beverages containing sugar and milk (24 kcal/can), as well as high levels of CGA, reduces visceral fat area (VFA), body weight, and WC compared with placebo pre-packaged caffeinated coffee beverages (22 kcal/can) containing no CGA [26]. The most favorable form for consuming CGA, such as through spray-dried coffee (instant coffee), drip-coffee, or pre-packed coffee beverage; the most favorable source of CGA, such as green coffee beans or roasted coffee beans; and the most favorable roasting method, such as dark roasting or light roasting, however, are unclear. Moreover, the bioavailability of CGA may be modified as a consequence of interactions with food macronutrients or the food matrix [27]. Previous clinical interventional studies of the components of roasted or green coffee beans were controversial and had limitations, such as being unblinded and not placebo-controlled or having a small sample size and short duration. To our knowledge, no large-scale randomized clinical trials were conducted comparing instant, spray-dried coffee beverages with high levels of CGA and ordinary levels of caffeine with conventional spray-dried coffee beverages with low levels of CGA and ordinary levels of caffeine on abdominal fat accumulation. We hypothesized that CGA-containing coffee reduces abdominal fat area compared with common coffee beverages, regardless of the beverage form. The present study compared the effects of the consumption of instant coffee with high CGA levels with that of conventional instant coffee with low CGA levels over the course of 12 weeks on the change in abdominal VFA, as the primary endpoint in overweight individuals, and safety in healthy overweight adult men and women (body mass index (BMI) ≥25 to <30 kg/m^2^; class 1 obesity according to Japanese criteria).

## 2. Materials and Methods 

### 2.1. Subjects

Subjects were recruited from among men and women residing in the Hokkaido area in Japan. The inclusion criteria were as follows: BMI ≥ 25 to <30 kg/m^2^; VFA ≥ 80 cm^2^; age 20 to <65 years. Exclusion criteria were as follows: allergy to drugs or food; current disease or history of severe disease related to the liver, kidney, heart, lungs, or digestive system; systolic blood pressure <90 mmHg or ≥160 mmHg; heavy drinking (>30 g alcohol/day) or smoking; extremely irregular dietary habits; and shift work or late-night work. The sample size calculation was based on that of a previous trial assessing the effects of CGA-containing pre-packed coffee beverages on overweight subjects [26]. The number of subjects (71 subjects/group) was calculated on the basis of the assumption of a change in VFA (ΔVFA) of 5 cm^2^, standard deviation of 15 cm^2^, significance level of α = 0.05, and power (1 − β) = 0.80. To account for the potential for dropouts, the target number of subjects was set at 150. This trial was registered at www.umin.ac.jp/ctr/ as UMIN000036011. The present trial was approved by the Institutional Review Board of Miyawaki Orthopedics Clinic (Hokkaido, Japan) and conducted at Fukuhara Clinic (Hokkaido, Japan) in compliance with the ethical principles of the Declaration of Helsinki (2013) and trial protocol. The principal investigator explained the study purpose, details, and potential risks to each subject, both in writing and verbally, and obtained written informed consent from the subjects. The New Drug Development Research Center, Inc. (Hokkaido, Japan) was commissioned to carry out all tasks related to the trial, including subject recruitment, randomization, blinding, and statistical analyses. The trial was conducted from January 2014 to September 2014.

### 2.2. Materials

The CGA coffee was prepared from coffee beans using the general methods for preparing regular coffee but was modified to contain 300 mg of CGA as a minimum guaranteed amount per serving and to decrease the oxidant components through adsorptive treatment with activated carbon [28,29]. Instant coffee was prepared after spray-drying (Kao Corporation, Tokyo, Japan), and the resulting powder, dissolved in 180 mL of hot water, was defined as the CGA coffee and contained 369 mg of CGA/serving. The control coffee was prepared in the same manner as regular instant coffee and contained 35 mg of CGA/serving. The appearance of the CGA and control coffee beverages was indistinguishable. Table 1 shows the nutritional composition of the CGA coffee and control coffee.

### 2.3. Study Design

This randomized, double-blinded, parallel between-group comparison trial comprised a four-week pre-observation period, a 12-week treatment period, and a four-week post-observation period (16 weeks). Pre- and post-observation periods enable control for pre-existing trends and evaluation of the impact of an intervention over time. Subjects who met the inclusion criteria on the basis of the baseline examinations were selected, resulting in a final study group of 150 healthy men and women. At zero weeks, the subjects were allocated to two groups: a CGA coffee-consuming group and a control coffee-consuming group, through stratified randomization with the stratification factors of sex, age, and VFA. Examinations were performed at zero weeks and at four, eight, 12, and 16 weeks after the intervention. During the intervention period, subjects were instructed to consume the test drink once per day (180 mL/day) over the 12 weeks. The primary endpoint was VFA, and secondary outcomes included subcutaneous fat area (SFA), total abdominal fat area (TFA), BMI, and WC at 12 weeks, which were compared with the corresponding baseline values. During the entire trial period, the subjects were instructed not to take medications that influence body fat, foods anticipated to have body fat-reducing effects, or conventional coffee. The subjects were also instructed to avoid excessive eating and drinking, to maintain their normal dietary habits, and to avoid changing their exercise and smoking habits. The subjects were asked to refrain from drinking alcohol on the day before the examination, to finish supper before 9:00 p.m., and to avoid smoking and food and drinks other than water until the examination was completed.

### 2.4. Measurements

Subject height was measured only in the pre-trial observation period. Body weight and WC were measured at each visit. WC was measured at the level of the umbilicus in the standing position. BMI was calculated on the basis of body weight and height. Blood pressure was measured in the morning during each visit, and systolic blood pressure and diastolic blood pressure were measured once in a sitting position with an electronic sphygmomanometer after a resting period of at least 10 min. The abdominal fat area measurements were performed according to the method described by Tokunaga et al. [30]. Specifically, a computed tomography scanning system (CT-W450, Hitachi Healthcare Systems, Inc., Tokyo, Japan) was used to obtain umbilical region cross-sectional images at baseline, and at eight and 12 weeks. The imaging settings were as follows: tube voltage, 120 kVp; mAs value, 90 mAs; window level, 0; and window width, 1000. For film photographs, visceral fat measurement software (Fat Scan^TM^ Ver. 3.0, N2 System, Inc., Ibaraki, Japan) was used to calculate VFA, TFA, and SFA. Blood was drawn and urine samples were obtained from subjects after at least a 12-h fast at baseline, and at four, eight, and 12 weeks during the intervention, as well as during the post-intervention period. The blood parameters assessed included white blood cell count, red blood cell count, hemoglobin, hematocrit, platelet count, mean corpuscular volume (MCV), mean corpuscular hemoglobin, and mean corpuscular hemoglobin concentration. Blood was collected into tubes containing EDTA-2Na and centrifuged at 2000× *g* for 15 min at 4 °C to obtain plasma samples for HbA1c and glucose measurements. For measurements of total protein, albumin, total bilirubin, aspartate aminotransferase, alanine aminotransferase, lactate dehydrogenase, alkaline phosphatase, γ-glutamyl transferase, total cholesterol, high-density lipoprotein-cholesterol, low-density lipoprotein-cholesterol, triglyceride, uric acid, urea nitrogen, creatinine, sodium, chloride, and potassium, the blood samples were kept at room temperature for 15 min, and serum samples were obtained by centrifuging at 3000 rpm for 15 min at 4 °C. For urine samples, the following parameters were measured: specific gravity, pH, ketone bodies, occult blood reaction, urobilinogen, bilirubin, protein, and glucose. All blood and urine measurements were performed by Daiichi Kishimoto Clinical Laboratories, Inc. (Hokkaido, Japan).

### 2.5. Dietary Record and Physical Activity

The daily diet and amounts of physical activity were assessed for the three days before the zero-week and twelve-week examinations performed by registered dietitians. Subjects recorded the details of their diet on a dietary recording form for 3 days and took pictures of all meals, to show that eating habits and exercise life did not change. Food content was analyzed on the basis of both the diet form and pictures by a dietician using Excel-Eiyou-kun^TM^ Ver. 6 software (Kenpakusha, Tokyo, Japan), and the total calories and carbohydrate, fat, and protein intake were calculated. Moreover, starting three days before the zero-week and 12-week examinations, the subjects wore a pedometer (TANITA Corporation, Tokyo, Japan) and recorded their daily number of steps.

### 2.6. Adverse Events

Adverse events were evaluated by the principal physician on the basis of diaries kept by the subjects during the trial period, as well as interviews with subjects on examination day visits. Every day over the course of the 16 weeks spanning the intervention and post-intervention periods, the subjects recorded the times at which the test drink was consumed, as well as any subjective symptoms. Interviews by the principal investigator were conducted every four weeks during hospital visits. During these interviews, if subjects noted the appearance of adverse events, they were asked to record details, such as the type of symptom, time at which the symptom appeared, degree of symptom, measures taken, outcomes of the measures, and relationship of the symptom to the test drink in the case report form, and the causal relationship between the adverse events and test drink consumption was determined.

### 2.7. Statistical Analysis

All values are presented as mean ± standard deviation. Actual measured values are shown, and, for VFA, SFA, TFA, body weight, BMI, and WC, values at 12 weeks relative to those at zero weeks or baseline are also provided. The amounts of energy intake, protein, fat, carbohydrate, and daily steps at zero weeks and 12 weeks are presented as mean values calculated from measurements from three consecutive days. These mean values were used in subsequent analyses. For primary endpoint evaluation, Student’s *t*-test was used to test for significant differences when the data were normally distributed. For primary endpoint evaluation, we did not correct for multiplicity in tests for each time point, because the primary endpoint was initially set as the change at 12 weeks relative to baseline. We also performed repeated-measures ANOVA using a linear mixed model with fixed effects of test drink and time as main effects, and test drink × time as the interaction. The MIXED procedure (SAS) was used for linear mixed models, and the significance of the fixed effect was estimated using robust variance. The number of adverse events was assessed with Fisher’s exact test in all subjects. Statistical analyses were performed with SAS ver. 9.3 (SAS Institute, Cary, NC, USA). The significance level was set at 5% (two-sided), and *p* < 0.05 was considered statistically significant.

## 3. Results

### 3.1. Subjects

Figure 1 shows the study flow from subject selection to analysis. From a total of 284 individuals who underwent screening, 150 subjects who met the inclusion criteria and not the exclusion criteria were selected and randomly allocated to the control group or CGA group (*n* = 75/group). Five subjects in the control group and three subjects in the CGA group withdrew consent before the intervention period. Accordingly, 142 subjects (control group, 70; CGA group, 72) participated in and completed the trial. The subject characteristics are summarized in Table 2. The test drink consumption rates were 99.8% ± 0.5% in the control group and 99.6% ± 0.8% in the CGA group, with no significant difference between the two groups. Table 3 shows the energy intake, protein intake, fat intake, carbohydrate intake, and physical activity (number of steps) at zero and 12 weeks. There was no group × time interaction, and no change in dietary content or physical activity during the test drink consumption period.

### 3.2. Abdominal Fat Area and Anthropometric Measurements

The absolute values and changes from baseline in VFA, SFA, and TFA are shown in Table 4. The CGA group exhibited a significant decrease relative to the control group at 12 weeks by *t*-test in VFA (CGA group: −9.0 cm^2^, control group: −1.0 cm^2^, *p* = 0.025) and changes in VFA from baseline, i.e., ΔVFA (*p* < 0.001). The VFA and ΔVFA in the CGA group were significantly different with a group × time interaction (*p* < 0.001) and a group effect (*p* < 0.001), compared with the control group by repeated-measures ANOVA. The change in the TFA from baseline (ΔTFA) was significantly smaller in the CGA group compared with the control group at 12 weeks (*p* < 0.001, *t*-test). The TFA and ΔTFA in the CGA group showed a significant group × time interaction (*p* = 0.001) and group effect (*p* < 0.001), compared with the control group by repeated-measures ANOVA. In contrast, neither the SFA nor the ΔSFA differed significantly between the two groups at 12 weeks. The absolute values and changes in body weight, BMI, WC, systolic blood pressure, and diastolic blood pressure are shown in Table 5. Repeated-measures ANOVA revealed a group × time interaction for body weight, BMI, and WC (*p* = 0.025, *p* = 0.015, *p* = 0.001, respectively). The Δweight, ΔBMI, and ΔWC in the CGA group had a significant group × time interaction compared with the control group by repeated-measures ANOVA (*p* = 0.010, *p* = 0.006, and *p* = 0.012, respectively). Moreover, a significant group effect was detected for ΔWC (*p* = 0.023); i.e., WC in the CGA group was significantly decreased (0.8 cm) relative to that in the control group at 12 weeks (*p* = 0.001, *t*-test). With respect to Δweight and ΔBMI, the CGA group showed a decreasing trend relative to the control group (*p* = 0.080 and *p* = 0.071, respectively).

### 3.3. Blood and Urine Tests

The blood test parameters are shown in Table 6. The changes in potassium showed a significant group × time interaction (*p* = 0.025) and a significant group effect (*p* = 0.012). No group × time interaction was noted for the other assessed parameters. Although a significant group effect was detected for MCV and γ-glutamyl transferase (*p* = 0.045 and *p* = 0.005, respectively), the fluctuations were within the reference range. A significant effect of time was observed for the following: red blood cell count, hemoglobin, hematocrit, MCV, total protein, aspartate aminotransferase, alanine aminotransferase, alkaline phosphatase, HbA1c, uric acid, creatinine, sodium, and chloride. The urinalysis revealed no significant changes in specific gravity, pH, ketone bodies, occult blood reaction, urobilinogen, bilirubin, protein, and glucose; moreover, no clinically problematic fluctuations were observed.

### 3.4. Adverse Events

Adverse events were determined from diaries kept by the subjects during the trial period and physician interviews. In total, 72 adverse events were noted in 25 subjects from the control group, including headache (*n* = 8), sore throat (*n* = 9), cough (*n* = 9), runny nose/nasal congestion (*n* = 16), fatigue (*n* = 1), sneezing (*n* = 4), fever (*n* = 3), chills (*n* = 4), stomach ache (*n* = 3), lethargy/malaise (*n* = 3), drowsiness (*n* = 1), joint pain (*n* = 2), stomach discomfort (*n* = 1), phlegm (*n* = 3), toothache (*n* = 1), stiff shoulder (*n* = 1), lower back pain (*n* = 2), and itchy hands (*n* = 1). In total, 63 adverse events were noted in 29 subjects from the CGA group, including headache (*n* = 10), sore throat (*n* = 10), cough (*n* = 10), runny nose/nasal congestion (*n* = 12), fatigue (*n* = 1), sneezing (*n* = 3), fever (*n* = 5), chills (*n* = 1), stomach ache (*n* = 1), lethargy/malaise (*n* = 1), joint pain (*n* = 1), stomach discomfort (*n* = 1), loose bowels/diarrhea (*n* = 2), shoulder pain (*n* = 1), vomiting/nausea (*n* = 2), dizziness (*n* = 1), and nosebleed (*n* = 1). These symptoms were considered by the principal investigator to be mild and temporary, and to not have been caused by the test drink according to the symptoms, symptom duration and degree, and outcomes. The number of adverse events did not differ significantly between the two groups.

## 4. Discussion

In this study, we assessed the effectiveness of daily consumption of instant coffee containing high CGA levels and decreased oxidant capacity over the course of 12 weeks in healthy overweight adult men and women (BMI ≥ 25 to <30 kg/m^2^) for reducing abdominal fat area, with VFA as the primary endpoint measure. A significant decrease in VFA was detected in cross-sectional CT images. The secondary outcomes were TFA, body weight, BMI, and WC. Furthermore, according to the results of the diaries of subjects and physician interviews during the trial, there were no adverse events from drinking instant coffee containing high CGA levels. In a review article describing the effects of coffee on the risk of diabetes, Greenberg et al. [31] suggested that caffeine and CGA have a body weight-reducing effect. Previous studies reported benefits related to an increased CGA content, such as reductions in glucose absorption, body weight, body fat, and DNA damage [25,32]. Few randomized control trial studies, however, described the influence of CGA on body fat or abdominal visceral fat accumulation. In the present trial, compared with the control group, the CGA group showed significant decreases in VFA at 12 weeks relative to baseline. Similar results were observed for TFA. Moreover, group × time interactions were observed for BMI, body weight, and WC, which showed significant decreases relative to the control group. WC is highly correlated with VFA [31], and the findings that not only VFA, but also BMI, body weight, and WC, were significantly decreased suggest that CGA coffee has a body fat-reducing effect. The changes in body weight, BMI, and WC in the CGA group tended to be smaller than that in previous studies [20,26]. This discrepancy may be related to the seasons during which the intervention periods occurred and the daily conditions of the subjects. Further comparison studies are needed. Food intake and physical activity during the trial did not differ significantly between groups. The CGA coffee and control coffee were both instant black coffee with essentially the same amounts of caffeine and calories. Because the only difference between the components of each drink was the amount of CGA, these results suggest that CGA was responsible for the decreases in VFA, BMI, body weight, and WC. Although no studies clearly demonstrated a relationship between oxidative stress and abdominal visceral fat, one study reported that oxidant components enhance the energy expenditure and fat oxidation effects of CGA-containing coffee [33]. The mechanism underlying the effect of instant coffee containing high amounts of CGA and decreased amounts of oxidant components on decreasing abdominal visceral fat might involve both increased energy expenditure and increased fat burning. According to a human clinical trial comparing daily consumption of roasted coffee containing 359 mg CGA and placebo coffee for one week, repeated consumption of CGA decreased the respiratory quotient and increased oxygen consumption [34], which means that both energy expenditure and fat oxidation were increased. Furthermore, feeding green coffee bean extract containing CGA to mice with diet-induced obesity affects body fat accumulation, with a dose-dependent inhibition of the increase in body weight and accumulation of visceral fat and liver fat [35]. In addition, analysis of the expression of energy metabolism-related genes in the liver revealed significantly decreased messenger RNA (mRNA) expression of stearoyl-CoA desaturase 1, acetyl-CoA carboxylase 1 (ACC1), and ACC2, which are involved in fatty-acid oxidation, after mice were administered green coffee bean extract containing CGA for two weeks. Decreased expression of stearoyl-CoA desaturase 1 and ACC1, which are involved in fatty-acid synthesis, promotes energy expenditure, and decreased ACC2 expression increases fatty-acid oxidation in the mitochondria through the reduced production of malonyl CoA, which inhibits carnitine palmitoyltransferase 1 [36]. Although CGA reportedly has a hypotensive effect [17,27,37,38], no significant changes in blood pressure were observed in the present trial. Because we did not target a population with blood pressure classified as greater or equal to stage I hypertension, as in previous trials studying the hypotensive effect of CGA, we were unable to observe any blood pressure-lowering effects. In terms of safety, although we found changes in the blood concentrations of potassium, the fluctuations were within the reference range (3.5–5.3 mEg/L) and were, thus, unlikely to be clinically problematic. Moreover, urinalysis revealed no problematic changes, and no adverse events considered to be due to the test drink were reported. This study had some limitations. Firstly, our subjects were Japanese; thus, whether the same effects might occur in other populations (e.g., Western populations) remains unclear. Secondly, although CGA was the effective component in the coffee used in this trial, blood CGA concentrations were not measured. Thus, the relationships between the blood CGA concentration and the outcome measures (e.g., VFA) are unclear. Extending the test period might have revealed a clearer difference in the effect.

## 5. Conclusions

Daily consumption of instant coffee containing high amounts of CGA significantly decreased VFA, TFA, BMI, body weight, and WC, without severe adverse events.

## Figures and Tables

**Figure 1 nutrients-11-01617-f001:**
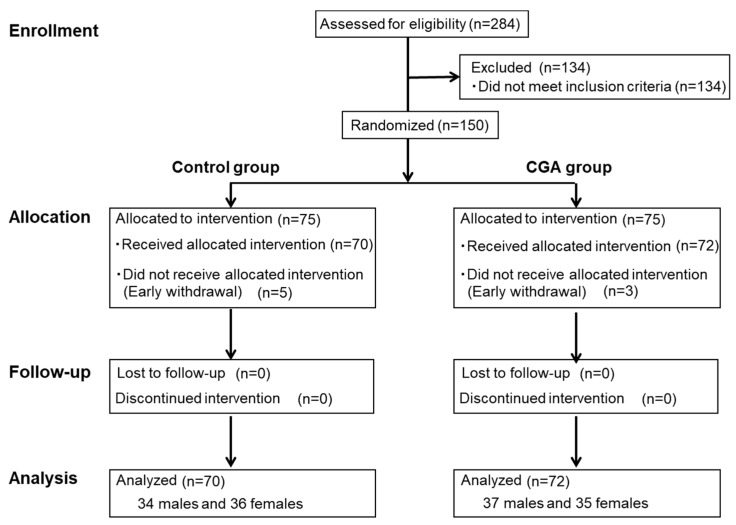
Flow diagram of subject selection and analysis.

**Table 1 nutrients-11-01617-t001:** Composition and nutritional content of test coffee. CGA—chlorogenic acid.

	Control Group	CGA Group
Composition		
Chlorogenic acid (mg) six analogues	30	319
Chlorogenic acid (mg) nine analogues	35	369
Caffeine (mg)	38	37
Hydroxyhydroquinone (mg)	0.858	0.054
Nutritional content		
Energy (kcal)	5	14
Protein (g)	0.28	0.70
Carbohydrate (g)	0.97	2.83
Fat (g)	0	0.61

Serving size: 180 mL. Chlorogenic acid (sixanalogues) analogues present are caffeoylquinic acids (3-CQA, 4-CQA, and 5-CQA) and feruloylquinic acids (3-CQA, 4-CQA, and 5-CQA). Chlorogenic acid (nineanalogues) analogues present are caffeoylquinic acids (3-CQA, 4-CQA, and 5-CQA), feruloylquinic acids (3-CQA, 4-CQA, and 5-CQA), and dicaffeoylquinic acids (3,4-di-CQA, 3,5-di-CQA, and 4,5-di-CQA).

**Table 2 nutrients-11-01617-t002:** Baseline characteristics of subjects.

	Control Group	CGA Group
	(*n* = 70)	(*n* = 72)
Age (years)	49.5 ± 8.4	49.8 ± 8.0
Height (cm)	162.0 ± 9.2	162.9 ± 9.7
Body weight (kg)	72.0 ± 9.1	72.5 ± 8.5
BMI (kg/m^2^)	27.3 ± 1.4	27.3 ± 1.4
SBP (mmHg)	135.9 ± 14.4	132.1 ± 13.1
DBP (mmHg)	84.0 ± 10.1	80.6 ± 9.2 *

Data are presented as mean ± SD. BMI: body mass index, SBP: systolic blood pressure, DBP: diastolic blood pressure; * *p* < 0.05, compared with the control group.

**Table 3 nutrients-11-01617-t003:** Dietary intake at 0 and 12 weeks in control and CGA groups.

		0 Weeks	12 Weeks	*p*-Value		
				Group	Time	Group × Time
Energy intake (kcal)						
	Control	1884.6 ± 331.1	1831.2 ± 365.5	0.540	0.030	0.760
	CGA	1935.4 ± 421.1	1864.5 ± 408.1			
Protein (g)						
	Control	69.3 ± 15.3	66.6 ± 15.6	0.332	0.052	0.860
	CGA	72.3 ± 20.5	69.1 ± 19.8			
Fat (g)						
	Control	62.2 ± 16.5	61.3 ± 18.4	0.572	0.288	0.699
	CGA	64.9 ± 19.8	62.5 ± 22.0			
Carbohydrate (g)						
	Control	248.9 ± 60.7	239.0 ± 52.5	0.668	0.069	0.636
	CGA	251.9 ± 61.9	245.2 ± 51.9			
Exercise (Steps)						
	Control	7323.0 ± 3910.1	7280.1 ± 3745.4	0.552	0.613	0.503
	CGA	7400.8 ± 3864.9	7740.4 ± 3420.6			

Data are presented as mean ± SD. Control (*n* = 70), CGA (*n* = 72). Group represents *p*-value in effect of group by repeated-measures ANOVA. Time represents *p*-value in effect of time by repeated-measures ANOVA. Group × time represents *p*-value in effect of group × time interaction by repeated-measures ANOVA.

**Table 4 nutrients-11-01617-t004:** Changes in physical assessment parameters in control and CGA groups.

		Baseline	8 Weeks	12 Weeks	*p*-Value		
					Group	Time	Group × Time
VFA (cm^2^)	Control	117.9 ± 24.4	116.3 ± 26.4	116.9 ± 24.4	0.111	<0.001	<0.001
	CGA	116.6 ± 25.2	106.9 ± 25.7 *	107.6 ± 25.2 *			
ΔVFA (cm^2^)	Control		−1.6 ± 15.4	−1.0 ± 14.3	<0.001	0.541	0.811
	CGA		−9.7 ± 12.4 **	−9.0 ± 13.9 ***			
SFA (cm^2^)	Control	223.0 ± 58.4	223.1 ± 57.0	222.1 ± 58.7	0.584	0.030	0.279
	CGA	219.3 ± 52.7	216.2 ± 52.2	214.5 ± 54.9			
ΔSFA (cm^2^)	Control		0.1 ± 12.6	−0.9 ± 12.0	0.118	0.217	0.772
	CGA		−3.1 ± 14.5	−4.8 ± 14.9			
TFA (cm^2^)	Control	341.0 ± 53.9	339.4 ± 56.1	339.0 ± 58.0	0.203	<0.001	0.001
	CGA	335.9 ± 52.8	323.1 ± 54.1	322.1 ± 54.6			
ΔTFA (cm^2^)	Control		−1.6 ± 20.3	−2.0 ± 16.2	<0.001	0.717	0.713
	CGA		−12.8 ± 21.2 **	−13.8 ± 22.9 ***			

Data are presented as mean ± SD. Control (*n* = 70), CGA (*n* = 72). VFA: visceral fat area, SFA: subcutaneous fat area, TFA: total fat area. Group represents *p*-value calculated by repeated-measures ANOVA. Group × time represents *p*-value calculated for group × time interaction by repeated-measures ANOVA. * *p* < 0.05, ** *p* < 0.01, and *** *p* < 0.001, compared with the control group.

**Table 5 nutrients-11-01617-t005:** Changes in physical assessment parameters in control and CGA groups.

		0 Weeks	4 Weeks	8 Weeks	12 Weeks	Post-Intervention	*p*-Value		
							Group	Time	Group × Time
Body weight (kg)	Control	71.8 ± 9.1	71.9 ± 9.2	71.9 ± 9.2	71.9 ± 9.2	71.8 ± 9.2	0.809	0.005	0.025
	CGA	72.0 ± 8.5	72.3 ± 8.4	72.0 ± 8.5	71.8 ± 8.5	71.9 ± 8.5			
ΔBody weight (kg)	Control		0.1 ± 0.8	0.1 ± 1.0	0.1 ± 1.1	0.0 ± 1.3	0.592	0.008	0.010
	CGA		0.3 ± 0.9	0.0 ± 0.9	−0.2 ± 1.0	−0.1 ± 1.3			
BMI (kg/m^2^)	Control	27.3 ± 1.5	27.3 ± 1.5	27.3 ± 1.5	27.3 ± 1.6	27.3 ± 1.6	0.375	0.005	0.015
	CGA	27.1 ± 1.3	27.2 ± 1.4	27.1 ± 1.3	27.0 ± 1.3	27.0 ± 1.3			
ΔBMI (kg/m^2^)	Control		0.04 ± 0.03	0.05 ± 0.38	0.06 ± 0.42	0.02 ± 0.48	0.612	0.012	0.006
	CGA		0.12 ± 0.34	−0.01 ± 0.33	−0.07 ± 0.37	−0.04 ± 0.48			
Waist circumference (cm)	Control	92.8 ± 4.5	92.6 ± 4.5	92.5 ± 4.5	92.9 ± 4.6	92.7 ± 4.7	0.525	0.002	0.001
	CGA	92.7 ± 5.2	92.2 ± 5.1	92.0 ± 5.1	92.0 ± 5.0	92.3 ± 4.9			
ΔWaist circumference (cm)	Control		−0.2 ± 1.3	−0.3 ± 1.6	0.1 ± 1.6	−0.1 ± 2.0	0.023	0.065	0.012
	CGA		−0.5 ± 1.3	−0.7 ± 1.6	−0.7 ± 1.4 **	−0.4 ± 1.7			
SBP (mmHg)	Control	134.0 ± 12.0	133.7 ± 13.1	132.5 ± 13.3	134.5 ± 12.4	130.1 ± 14.9	0.520	0.326	0.812
	CGA	132.5 ± 13.5	132.4 ± 13.1	131.5 ± 13.0	132.5 ± 13.9	125.8 ± 12.4			
DBP (mmHg)	Control	82.7 ± 9.5	81.7 ± 10.0	81.4 ± 9.6	82.8 ± 9.5	78.9 ± 10.5	0.144	0.260	0.395
	CGA	80.7 ± 9.4	80.3 ± 8.5	79.3 ± 8.9	79.8 ± 9.5	75.5 ± 10.1			

Data are presented as mean ± SD. Control (*n* = 70), CGA (*n* = 72). BMI: body mass index, SBP: systolic blood pressure, DBP: diastolic blood pressure. Group represents *p*-value in effect of group by repeated-measures ANOVA. The *p*-values were calculated for group, time, and group × time interaction by repeated-measures ANOVA. ** *p* < 0.01, compared with the control group.

**Table 6 nutrients-11-01617-t006:** Changes in blood parameters in control and CGA groups.

		0 Weeks	4 Weeks	8 Weeks	12 Weeks	Post-Intervention	*p*-Value		
							Group	Time	Group × Time
White blood cell count (/μL)									
	Control	6673 ± 1771	6759 ± 1852	6773 ± 1567	6906 ± 1596	6870 ± 1660	0.893	0.211	0.583
	CGA	6858 ± 1558	6849 ± 1546	6701 ± 1476	6892 ± 1658	6732 ± 1599			
Red blood cell count (×10^4^/μL)									
	Control	478 ± 38	475 ± 37	475 ± 39	475 ± 39	471 ± 40	0.586	<0.001	0.218
	CGA	477 ± 39	469 ± 42	471 ± 40	471 ± 40	464 ± 41			
Hemoglobin (g/dL)									
	Control	14.3 ± 1.6	14.3 ± 1.6	14.2 ± 1.5	14.3 ± 1.5	14.1 ± 1.5	0.819	<0.001	0.204
	CGA	14.0 ± 1.3	14.2 ± 1.2	14.3 ± 1.2	14.3 ± 1.2	14.0 ± 1.3			
Hematocrit (%)									
	Control	42.8 ± 3.7	42.4 ± 3.5	42.3 ± 3.6	42.5 ± 3.6	42.7 ± 3.6	0.532	<0.001	0.219
	CGA	43.3 ± 3.1	42.5 ± 3.2	42.7 ± 3.2	43.0 ± 3.2	42.6 ± 3.4			
Platelets (×10^4^/μL)									
	Control	25.8 ± 5.6	26.0 ± 5.8	26.5 ± 6.1	26.0 ± 5.9	25.9 ± 5.6	0.355	0.339	0.282
	CGA	25.4 ± 5.2	25.3 ± 5.6	25.3 ± 6.5	25.2 ± 5.7	25.1 ± 5.4			
MCV (fL)									
	Control	89.7 ± 4.9	89.4 ± 4.8	89.2 ± 4.7	89.6 ± 4.2	90.6 ± 4.2	0.045	<0.001	0.466
	CGA	91.0 ± 4.4	90.8 ± 4.2	90.8 ± 4.3 *	91.3 ± 4.4 **	92.2 ± 4.7			
MCH (pg)									
	Control	30.0 ± 2.2	30.0 ± 2.2	29.9 ± 2.1	30.0 ± 1.9	29.9 ± 1.9	0.212	0.051	0.764
	CGA	30.4 ± 1.6	30.4 ± 1.6	30.4 ± 1.6	30.4 ± 1.6	30.3 ± 1.7			
MCHC (%)									
	Control	33.4 ± 1.2	33.5 ± 1.2	33.5 ± 1.2	33.5 ± 1.1	33.0 ± 1.1	0.504	0.070	0.251
	CGA	33.4 ± 0.8	33.5 ± 0.9	33.4 ± 0.8	33.3 ± 0.9	32.9 ± 0.9			
Total protein (g/dL)									
	Control	7.5 ± 0.4	7.4 ± 0.4	7.4 ± 0.4	7.3 ± 0.4	7.4 ± 0.5	0.051	<0.001	0.951
	CGA	7.4 ± 0.3	7.3 ± 0.3	7.3 ± 0.3	7.2 ± 0.3	7.2 ± 0.4			
Albumin (g/dL)									
	Control	4.5 ± 0.2	4.5 ± 0.3	4.5 ± 0.3	4.5 ± 0.3	4.4 ± 0.3	0.644	0.550	0.846
	CGA	4.5 ± 0.3	4.4 ± 0.3	4.5 ± 0.3	4.4 ± 0.3	4.4 ± 0.2			
Total bilirubin (mg/dL)									
	Control	0.66 ± 0.30	0.63 ± 0.25	0.62 ± 0.20	0.64 ± 0.24	0.66 ± 0.26	0.636	0.100	0.999
	CGA	0.68 ± 0.27	0.66 ± 0.27	0.65 ± 0.24	0.67 ± 0.26	0.67 ± 6.70			
AST (U/L)									
	Control	23.2 ± 8.6	22.2 ± 6.6	22.4 ± 7.2	22.6 ± 7.3	21.1 ± 6.3	0.214	0.042	0.990
	CGA	22.7 ± 10.0	21.5 ± 8.0	21.7 ± 9.6	22.6 ± 12.2	22.5 ± 10.3			
ALT (U/L)									
	Control	28.9 ± 18.7	27.0 ± 16.2	27.5 ± 16.5	29.9 ± 17.3	26.2 ± 13.5	0.164	<0.001	0.975
	CGA	26.3 ± 18.6	24.0 ± 14.8	25.0 ± 16.9	27.3 ± 17.0	26.3 ± 16.5			
LDH (U/L)									
	Control	175.0 ± 30.2	176.2 ± 31.4	177.1 ± 33.4	174.9 ± 33.9	180.6 ± 4.7	0.645	0.701	0.349
	CGA	175.1 ± 22.9	174.0 ± 28.6	173.1 ± 26.0	173.2 ± 28.8	182.2 ± 29.2			
ALP (U/L)									
	Control	223.6 ± 60.6	228.9 ± 61.2	224.4 ± 57.9	224.6 ± 62.7	227.3 ± 62.5	0.110	0.025	0.182
	CGA	212.5 ± 52.5	211.9 ± 56.5	207.6 ± 53.4	209.0 ± 57.6	208.0 ± 53.0			
γ-GT (U/L)									
	Control	38.1 ± 27.6	37.7 ± 23.9	37.3 ± 24.2	39.7 ± 28.6	39.5 ± 28.3	0.005	0.053	0.129
	CGA	30.6 ± 19.1	27.4 ± 14.3	27.8 ± 15.4	28.5 ± 16.3	29.6 ± 17.6			
Total cholesterol (mg/dL)									
	Control	218.4 ± 34.0	218.8 ± 33.7	218.2 ± 32.6	218.3 ± 35.3	217.3 ± 32.6	0.390	0.284	0.341
	CGA	220.8 ± 35.0	225.4 ± 38.0	224.8 ± 36.6	227.1 ± 35.8	216.7 ± 36.2			
HDL cholesterol (mg/dL)									
	Control	56.9 ± 12.7	56.0 ± 12.6	55.3 ± 11.5	56.2 ± 13.9	57.1 ± 13.0	0.332	0.163	0.666
	CGA	58.3 ± 13.7	57.7 ± 14.4	57.8 ± 13.7	58.4 ± 13.9	58.8 ± 13.3			
LDL cholesterol (mg/dL)									
	Control	141.4 ± 29.7	138.1 ± 27.5	139.3 ± 26.8	139.9 ± 29.6	136.5 ± 26.5	0.262	0.762	0.364
	CGA	145.0 ± 34.3	146.3 ± 35.8	146.2 ± 35.3	148.6 ± 33.5	139.0 ± 34.3			
Triglycerides (mg/dL)									
	Control	141.8 ± 73.9	153.7 ± 113.0	144.5 ± 74.1	145.3 ± 90.7	158.3 ± 104.4	0.087	0.324	0.965
	CGA	122.6 ± 69.3	132.5 ± 79.3	125.9 ± 63.3	128.8 ± 89.8	117.7 ± 62.0			
Fasting blood glucose (mg/dL)									
	Control	87.8 ± 7.7	88.8 ± 7.9	88.3 ± 10.1	89.3 ± 9.4	89.4 ± 7.9	0.488	0.059	0.545
	CGA	89.6 ± 10.2	90.2 ± 11.3	88.1 ± 9.2	89.9 ± 11.9	88.9 ± 8.5			
HbA1c (%)									
	Control	5.3 ± 0.4	5.4 ± 0.4	5.4 ± 0.4	5.4 ± 0.4	5.3 ± 0.4	0.499	<0.001	0.976
	CGA	5.4 ± 0.5	5.4 ± 0.5	5.5 ± 0.5	5.5 ± 0.5	5.4 ± 0.6			
Uric acid (mg/dL)									
	Control	5.6 ± 1.4	5.5 ± 1.4	5.4 ± 1.3	5.5 ± 1.3	5.5 ± 1.3	0.387	<0.001	0.063
	CGA	5.5 ± 1.3	5.3 ± 1.3	5.1 ± 1.3	5.4 ± 1.3	5.4 ± 1.3			
Urea nitrogen (mg/dL)									
	Control	12.8 ± 3.2	12.7 ± 3.4	12.9 ± 3.6	12.7 ± 3.2	13.9 ± 4.1	0.824	0.357	0.086
	CGA	12.7 ± 3.0	13.1 ± 2.9	12.4 ± 2.8	12.5 ± 2.8	13.6 ± 3.2			
Creatinine (mg/dL)									
	Control	0.70 ± 0.16	0.68 ± 0.15	0.72 ± 0.17	0.76 ± 0.15	0.76 ± 0.16	0.537	<0.001	0.827
	CGA	0.71 ± 0.16	0.70 ± 0.16	0.74 ± 0.17	0.78 ± 0.17	0.77 ± 0.17			
Sodium (mEq/L)									
	Control	140.1 ± 1.5	139.5 ± 2.0	140.8 ± 1.4	139.8 ± 1.5	141.9 ± 1.7	0.271	<0.001	0.834
	CGA	140.3 ± 1.5	139.9 ± 2.0	140.9 ± 1.4	140.0 ± 1.4	141.8 ± 1.6			
Chloride (mEq/L)									
	Control	102.9 ± 1.7	102.3 ± 2.0	103.4 ± 1.7	102.4 ± 1.9	104.4 ± 1.8	0.515	<0.001	0.528
	CGA	102.9 ± 2.0	102.7 ± 2.2	103.3 ± 1.9	102.5 ± 2.0	103.9 ± 1.9			
Potassium (mEq/L)									
	Control	4.1 ± 0.3	4.2 ± 0.3	4.2 ± 0.3	3.9 ± 0.3	3.7 ± 0.3	0.012	0.053	0.025
	CGA	4.1 ± 0.2	4.3 ± 0.4 *	4.3 ± 0.3 *	4.1 ± 0.3 **	3.7 ± 0.3			

Data are presented as mean ± SD. Control (*n* = 70), CGA (*n* = 72). MCV: mean corpuscular volume, MCH: mean corpuscular hemoglobin, MCHC: mean corpuscular hemoglobin concentration, AST: aspartic aminotransferase, ALT: alanine aminotransferase, LDH: lactate dehydrogenase, ALP: alkaline phosphatase, γ-GT: γ-glutamyltranspeptidase, HDL: high-density lipoprotein, LDL: low-density lipoprotein, HbA1c: hemoglobin A1c. The *p*-values were calculated for group, time, and group × time interaction by repeated-measures ANOVA. * *p* < 0.05 and ** *p* < 0.01, compared with the control group.

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
