# Peer review of "Coffee Abundant in Chlorogenic Acids Reduces Abdominal Fat in Overweight Adults: A Randomized, Double-Blind, Controlled Trial"

_nutrients, 2019, doi:10.3390/nu11071617_

Reviewer 1 Report

This study evaluates the role of the functional ingredient chlorogenic acids (CGA) in the management of central obesity, which may add some evidence to the growing field of functional foods in the management of obesity and related disorders. However, there are two major concerns: 1) authors have already conducted a very similar study [26], previously, and do not clearly highlight the need for this study; if to be published, this study should serve as a follow up to the first study, focusing on blood parameters as a means to explain the beneficial effects of CGA on abdominal fat in overweight individuals; and 2) authors concluded the safety of CGA based on subjective observations from participants on adverse events. Concluding the safety of an ingredient requires a combination of in vitro and in vivo studies, and cannot be based on findings from one study. Because of these two major concerns, I recommend the rejection of this article.

Here are some comments that may benefit the authors when reviewing/rewriting their manuscript:

- Abstract: Line 12, lacks a background sentence to support the need for this study. Line 17, what do authors mean by 4-week pre- and post-observation periods? Line 18, what type of blood chemistry was assessed? Lines 24-26, this concluding sentence is just a repetition of the results; it does not include any practical implications of these findings. Also, authors use the word "safe", please refer to my comment above regarding safety.

- Keywords: BMI should be spelled out. The focus of this study was on abdominal fat not body weight per se. Suggest to change this keyword. 

- Introduction: lines 38 and 39, sentence repeating the same concept twice about the antioxidant properties of CGA. Lines 46-48, how was CGA consumed? what were the participants' characteristics? Lines 52-54, not clear sentence; suggest to rewrite. Lines 57-59, this is the study already conducted by authors; very similar to the current study with no clear justification for the current study except that caffeine was not increased but kept at normal levels; authors need to make a stronger rationale for this study and focus more on the blood parameters as mechanistic explanations to the effects of CGA on abdominal obesity. Line 67, VFA used as an abbreviation without previous explanation. Line 68, authors refer to safety - please refer to my comment above. Line 68, BMI used without explanation of the abbreviation. Also, what were the hypotheses of this study? they need to be clearly stated in addition to the rationale. 

- Materials and Methods: Line 92, since there are many analog forms for CGA, which one was added to the coffee treatment? what do authors mean by active? Lines 100 and 101, authors need to clarify what kind of assessment took place in the 4-week pre- and post-intervention. Line 112, how did authors define excessive eating? Line 129, what does SFA? Some of the blood parameters measured are not relevant metabolically to abdominal obesity and related disorders such as white blood cells, red blood cells, etc. authors need to clarify why these measurements were performed instead of focusing on relevant parameters. Line 145, how did authors teach participants to perform 3-day food record? which aspects of the diet they were interested to analyze? why did authors assess physical activity? Line 152, the title of this section should be reconsidered based on my comment on safety, above.

- Results: Table 1, Lines 240-244, did authors measure all these analogs in the control and treatment coffee? 

- Discussion: line 278, authors refer to "various safety examinations"...what do they mean by that? Lines 303-304, "...CGA decreased the respiratory quotient and increased oxygen consumption"...what does that mean physiologically? Lines 310-313, all these physiological findings are stated without explaining their physiological meaning. Line 321, the word "safe" is used again - please check my comment above.

- Conclusions: this is just a repetition of findings. what does that mean from a practical perspective? also, what are the limitations of this study?

- References: good number and up-to-date. 

Author Response

--Our responses are blue. Please see the attachment."

This study evaluates the role of the functional ingredient chlorogenic acids (CGA) in the management of central obesity, which may add some evidence to the growing field of functional foods in the management of obesity and related disorders. However, there are two major concerns:

1) authors have already conducted a very similar study [26], previously, and do not clearly highlight the need for this study; if to be published, this study should serve as a follow up to the first study, focusing on blood parameters as a means to explain the beneficial effects of CGA on abdominal fat in overweight individuals;

--According to your comment, we have added sentences to the manuscript as requested (page 2 lines 58 to 67, 2 to74).

2) authors concluded the safety of CGA based on subjective observations from participants on adverse events. Concluding the safety of an ingredient requires a combination of in vitro and in vivo studies, and cannot be based on findings from one study. Because of these two major concerns, I recommend the rejection of this article.

--According to your comment, we have changed sentences to the manuscript as requested (page 4 lines 166, page 11 lines 287and292 to 294). We have deleted the sentence (page 12 lines337to338)

 Here are some comments that may benefit the authors when reviewing/rewriting their manuscript:

- Abstract: Line 12, lacks a background sentence to support the need for this study.

--According to your comment, we have added sentence to the abstract as requested (page 1 lines 12 and 13).

 Line 17, what do authors mean by 4-week pre- and post-observation periods?

--According to your comment, we have added sentence to the abstract as requested (page 3 lines 113 and 114) because there is word limitation of abstract.

 Line 18, what type of blood chemistry was assessed?

--According to your comment, we have deleted the sentence in the abstract.

 Lines 24-26, this concluding sentence is just a repetition of the results; it does not include any practical implications of these findings. Also, authors use the word "safe", please refer to my comment above regarding safety.

--According to your comment, we have changed sentences to abstract as requested (page 1 lines 24).

 - Keywords: BMI should be spelled out. The focus of this study was on abdominal fat not body weight per se. Suggest to change this keyword.

--According to your comment, we have changed key words (page 1 lines 27).

 - Introduction: lines 38 and 39, sentence repeating the same concept twice about the antioxidant properties of CGA.

--According to your comment, we have deleted the words (page 1 lines 40 and 41)

 Lines 46-48, how was CGA consumed? what were the participants' characteristics?

--We intended to show the CGA contents in Table 1 and subjects’ characteristics in Table 2 (Page 6 and 7).

 Lines 52-54, not clear sentence; suggest to rewrite. Lines 57-59, this is the study already conducted by authors; very similar to the current study with no clear justification for the current study except that caffeine was not increased but kept at normal levels; authors need to make a stronger rationale for this study and focus more on the blood parameters as mechanistic explanations to the effects of CGA on abdominal obesity.

--According to your comment, we have added sentences to the manuscript as requested (page 2 lines 58 to 67 and line 72 to74).

Line 67, VFA used as an abbreviation without previous explanation.

--We intended to show the VFA in previous lines (Page 1 lines 19).

 Line 68, authors refer to safety - please refer to my comment above.

--According to your comment, we have changed sentences to the manuscript as requested (page 4 lines 166, page 11 lines 287and292 to 294). We have deleted the sentence (page 12 lines337to338)

 Line 68, BMI used without explanation of the abbreviation.

--According to your comment, we have spelled out in the introduction as requested (page 2 lines 77).

 Also, what were the hypotheses of this study? they need to be clearly stated in addition to the rationale.

--According to your comment, we have added sentences to the manuscript as requested (page 2 lines 72 to 74).

 - Materials and Methods: Line 92, since there are many analog forms for CGA, which one was added to the coffee treatment? what do authors mean by active?

--According to your comment, we have deleted the words (page 3 lines 101)

 Lines 100 and 101, authors need to clarify what kind of assessment took place in the 4-week pre- and post-intervention.

--According to your comment, we have added sentence to the abstract as requested (page 3 lines 113 and 114).

 Line 112, how did authors define excessive eating?

--We intended to maintain their normal dietary habits (Page 3 lines 125).

 Line 129, what does SFA?

--We intended as subcutaneous fat area (Page 3 lines 121)

 Some of the blood parameters measured are not relevant metabolically to abdominal obesity and related disorders such as white blood cells, red blood cells, etc. authors need to clarify why these measurements were performed instead of focusing on relevant parameters.

--We intended to focus on safety assessments.

 Line 145, how did authors teach participants to perform 3-day food record?

--According to your comment, we have added words to the manuscript as requested (page 4 lines 158).

 which aspects of the diet they were interested to analyze? why did authors assess physical activity?

--According to your comment, we have added words to the manuscript as requested (page 4 lines 160,161).

 Line 152, the title of this section should be reconsidered based on my comment on safety, above.

--According to your comment, we have changed words in subtitle (page 4 lines 165).

 - Results: Table 1, Lines 240-244, did authors measure all these analogs in the control and treatment coffee?

--We have measured all analogs in the control and treatment coffee.

 - Discussion: line 278, authors refer to "various safety examinations"...what do they mean by that?

--According to your comment, we have changed sentences to discussion as requested (page 11 lines 287).

 Lines 303-304, "...CGA decreased the respiratory quotient and increased oxygen consumption"...what does that mean physiologically? Lines 310-313, all these physiological findings are stated without explaining their physiological meaning.

--According to your comment, we have changed sentences to discussion as requested (page 12 lines 320 and lines 325).

 Line 321, the word "safe" is used again - please check my comment above.

--According to your comment, we have deleted sentence to the manuscript as requested (page 12 lines 337,338).

 - Conclusions: this is just a repetition of findings. what does that mean from a practical perspective?

--According to your comment, we have changed sentences to discussion as requested (page 12 lines 341).

 also, what are the limitations of this study?

--According to your comment, we have added section (page 12 lines 342 to 346).

 - References: good number and up-to-date.

--According to your comment, we have up-date the references.

Reviewer 2 Report

The present paper compared the effects of daily consumption of instant coffee enriched chlorogenic acids (CGA) and control conventional coffee on abdominal fat areas.

Is a a randomized, double-blind, parallel controlled, trial, that included 150 healthy, overweight men and women.

Patients were randomly allocated to either the high-CGA coffee (369 mg 15 CGA/serving) or control coffee (35 mg CGA/serving) group

They concluded that consumption of instant high-CGA coffee for 12 weeks by overweight adults was safe and effectively lowered the VFA, TFA, BMI, and waist circumference..

The study is interesting.

The main concern I have is related to the short duration of follow up. The differences in physical assessment parameters are little even if  statistically significant.

The methodology is very good and the statistical analysis is adequate.

Introduction: can be shortened

Methods: adequate  

Results. Good

Discussion:  a “limitation of the study” need to be included. Also a comment on recent hypothesis that coffee could act as nutraceutical beverages can be included  (i.e. see doi: 10.2459/JCM.0000000000000626)

References: are good 

Author Response

"Please see the attachment." --Our responses are blue.

The present paper compared the effects of daily consumption of instant coffee enriched chlorogenic acids (CGA) and control conventional coffee on abdominal fat areas. Is a a randomized, double-blind, parallel controlled, trial, that included 150 healthy, overweight men and women. Patients were randomly allocated to either the high-CGA coffee (369 mg 15 CGA/serving) or control coffee (35 mg CGA/serving) group. They concluded that consumption of instant high-CGA coffee for 12 weeks by overweight adults was safe and effectively lowered the VFA, TFA, BMI, and waist circumference. The study is interesting. The main concern I have is related to the short duration of follow up. The differences in physical assessment parameters are little even if statistically significant. The methodology is very good and the statistical analysis is adequate. Introduction: can be shortened. Methods: adequate. Results. Good. Discussion:  a “limitation of the study” need to be included. Also a comment on recent hypothesis that coffee could act as nutraceutical beverages can be included (i.e. see doi: 10.2459/JCM.0000000000000626). References: are good.

--According to your comment, we have added limitation (page 12 lines 342 to 346).

Limitations:This study has some limitations. First, our subjects were Japanese, and thus whether the same effects might occur in other populations (e.g., Western populations) remains unclear. Second, although CGA was the effective component in the coffee used in this trial, blood CGA concentrations were not measured. Thus, the relationships between the blood CGA concentration and the outcome measures (e.g., VFA) are unclear.Extending the test period,might have revealeda clearer difference in the effect.

Reviewer 3 Report

I have provided my comments in the attached PDF.

Minor Comments:

Line 53: Could you bring some molecular mechanisms by which CGA may reduce body fat?

Line 54: Are there studies that show adverse effects of CGA on body? If so, it would be worthy to be mentioned.

Line 76: What about high blood pressure? Did you also consider blood pressure?

Line 106: Why you did not have a control group without drinking coffee in your study? Then you could see the effect of regular coffee.

Line 151: What did you assess in pre-trial period except height? Did you control their coffee consumption?

Line 182: Please add the figure number.

Line 296: There was also difference in macro nutrients. Did you run a test to see if these differences are significant?

Author Response

--Our responses are blue.

Line 53: Could you bring some molecular mechanisms by which CGA may reduce body fat?

--We intended to show the mechanisms in discussion (Page 12 lines 317 to 330).

Line 54: Are there studies that show adverse effects of CGA on body? If so, it would be worthy to be mentioned.

--We intended to show the adverse effects in result section (Page 6 lines 236 to 250), and principal investigator judged to not have been caused by the test drink.

Line 76: What about high blood pressure? Did you also consider blood pressure?

--We intended to consider about the effects of blood pressure (Page 12 lines 330 to 333).

Line 106: Why you did not have a control group without drinking coffee in your study? Then you could see the effect of regular coffee.

--Certainly, as you say, if you set the observation group, you might have been able to verify the effect of regular coffee. This time, only the comparison between the two groups was conducted because of the scale of the study.

Line 151: What did you assess in pre-trial period except height? Did you control their coffee consumption?

--Height was measured only at screening. During the intervention period, subjects were instructed to consume the test drink once per day (180 mL/day) over the 12 weeks. During the entire trial period, the subjects were instructed not to take conventional coffee.

Line 182: Please add the figure number.

--According to your comment, we have added figure number (page 5 lines 194).

Line 296: There was also difference in macro nutrients. Did you run a test to see if these differences are significant?

--We showed the energy intake, protein intake, fat intake, carbohydrate intake, and physical activity (number of steps) at 0 and 12 weeks in Table 3. There was no group × time interaction, and no change in dietary content or physical activity during the test drink consumption period.

Round  2

Reviewer 1 Report

All comments were addressed.

Only one comment still needs to be addressed: limitations should be added to the Discussion and not to the Conclusions section.

Some English editing is though still needed throughout the manuscript. 

Author Response

--Our responses are blue."Please see the attachment."

Only one comment still needs to be addressed: limitations should be added to the Discussion and not to the Conclusions section.

--According to your comment, we have moved limitations to the end of Discussion section. (page 13 lines 337 to 342).

Some English editing is though still needed throughout the manuscript.

--According to your comment, we asked native English-speaking person to correct the errors again and provide you with a certificate.

10 July 2019
To Whom It May Concern:
This letter is to certify that SciTechEdit International, LLC copy-edited a version (July,
2019) of the manuscript titled “Coffee abundant in chlorogenic acids reduces
abdominal fat in overweight adults: a randomized, double-blind, controlled trial”
by Takuya Watanabe, Shinichi Kobayashi, Tohru Yamaguchi, Masanobu Hibi, Ikuo
Fukuhara, and Noriko Osaki.
SciTechEdit International takes full responsibility for the processing and correction of
documents while in our hands. Implementation of the editorial changes suggested by
SciTechEdit International, LLC and changes made after the edited manuscript was
returned to the authors are at the sole discretion of the authors.
Please direct any questions regarding the original editing of this manuscript to me at the
email address below.
Sincerely,

Karin Mesches, PhD
President, SciTechEdit International
karin.mesches@scitechedit.com
